# New Standardized Procedure to Extract Glyphosate and Aminomethylphosphonic Acid from Different Matrices: A Kit for HPLC-UV Detection

**DOI:** 10.3390/jox15010023

**Published:** 2025-02-02

**Authors:** Francesco Chiara, Sarah Allegra, Elisa Arrigo, Daniela Di Grazia, Francesco Maximillian Anthony Shelton Agar, Raluca Elena Abalai, Sara Gilardi, Silvia De Francia, Daniele Mancardi

**Affiliations:** 1Department of Physics, University of Trento, Via Sommarive 14, 38123 Povo, Turin, Italy; francesco.chiara@unito.it; 2Department of Clinical and Biological Sciences, University of Torino, San Luigi Gonzaga Hospital Regione Gonzole 10, 10043 Orbassano, Turin, Italy; elisa.arrigo@unito.it (E.A.); daniela.digrazia@unito.it (D.D.G.); francescomaximilliananthony.sheltonagar@unito.it (F.M.A.S.A.); raluca.abalai@unito.it (R.E.A.); sara.gilardi@ircc.it (S.G.); silvia.defrancia@unito.it (S.D.F.); daniele.mancardi@unito.it (D.M.)

**Keywords:** glyphosate, AMPA, biological matrix, liquid chromatography, HPLC-UV

## Abstract

Background: Glyphosate has been extensively used as herbicide since the early 1970s. The daily exposure limit is set at 0.3 mg/kg bw/d in Europe and 1.75 mg/kg bw/d in the USA. Among its derivatives, aminomethylphosphonic acid is the most stable and abundant. Understanding their biological effects then requires reliable methods for quantification in biological samples. Methods: We developed and validated a fast, low-cost, and reliable chromatographic method for determining glyphosate and aminomethylphosphonic acid concentrations. The validation included following parameters: specificity, selectivity, matrix effect, accuracy, precision, calibration performance, limit of quantification, recovery, and stability. Sample extraction employed an anion exchange resin with elution using hydrochloric acid 50.0 mmol/L. For HPLC analysis, analytes were derivatized, separated on a C18 column with a mobile phase of phosphate buffer (0.20 mol/L, pH 3.0) and acetonitrile (85:15), and detected at 240 nm. Results: The method demonstrated high reliability and reproducibility across various matrices. Its performance met all validation criteria, confirming its suitability for quantifying glyphosate and aminomethylphosphonic acid in different biological and experimental setups. Conclusions: This method can offer a practical resource for applications in experimental research, medical diagnostics, quality control, and food safety.

## 1. Introduction

N-(phosphonomethyl)glycine (glyphosate, GLY) was synthesized for the first time in 1974 and used as herbicide for commercial distribution because of its selective inhibition of the shikimate pathways, a pivotal enzymatic pathway supporting plant growth [1]. Considered safe in terms of environmental impact and toxic effects on animals, with low production costs and high market demand, GLY rapidly reached a worldwide diffusion for extensive agricultural use and it has been the most adopted herbicide over the last five decades [2]. GLY is transformed in aminomethylphosphonic acid (AMPA) by plants and in soil, through oxidative deamination and further degradation to 2-methylphosphinicoacetic acid [1]. Since the first introduction to the market, the effects of GLY on plants have been largely investigated, leading to the development of genetically modified crops, resistant to the herbicide’s action [3,4].

Unlike fungi, plants, algae and bacteria, animals retrieve shikimate-derived aromatic amino acids through their diet and lack the target enzyme. Nevertheless, the demonstrated effects on microbiota can indirectly account for deleterious outcomes on mammalian physiological processes [1]. Scientific literature about the biological effects of GLY on humans is still controversial and lacks consensus among research groups and certifying agencies [5,6,7]. In this context, establishing a consistent and transversal way to extract, determine, and quantify GLY and AMPA in biological samples is of pivotal importance to address potential pharmacodynamic and pharmacokinetics features of the two molecules [8]. Most of the described procedures to determine concentrations, however, are extremely complicated, time-consuming, and lack acceptable quantification [9,10]. A robust literature review is essential to position our method within the current research context. Numerous studies have addressed the determination of GLY and AMPA using techniques which include FMOC-based derivatizations, liquid chromatography coupled with mass spectrometry (LC-MS/MS), and fluorometric methods. However, many of these approaches have significant limitations. For instance, the FMOC derivatization protocol, widely employed, is complex, requires extended preparation times, and is often not cross-applicable to various biological matrices. Additionally, LC-MS/MS-based methods, while extremely sensitive and specific, require expensive instrumentation and highly trained personnel, limiting their applicability in routine laboratories.

Recent research highlights the significance of glyphosate and its primary metabolites in environmental and human exposure contexts. Buekers et al. [11] examined glyphosate and AMPA levels in human urine across multiple European regions, reporting concentrations below regulatory thresholds, yet emphasizing the need for subgroup-specific data (e.g., occupational exposure) to refine risk assessments. Similarly, Geerdink et al. [12] developed an advanced ion chromatography method for glyphosate detection in surface water, revealing widespread environmental contamination and underscoring the importance of method sensitivity for regulatory compliance. Lastly, Souza et al. [13] reported high glyphosate and AMPA concentrations in Brazilian hydrographic basins, highlighting regional differences in agricultural practices and contamination levels. These findings underscore the importance of continued monitoring and method development to address public and environmental health concerns.

Our approach, based on derivatization with 4-Toluenesulfonyl chloride, a method previously applied by Kawai S. [14], Tomita M. [15], and Khrolenko M.V. [16], was designed to overcome these challenges. Compared to FMOC, our method is simpler, faster, and more adaptable to a broader range of matrices. The ease of derivatization and compatibility with widely available HPLC-UV systems amplify its practical utility. This innovation significantly reduces operational costs while maintaining high standards of accuracy and precision.

In some cases, researchers attempting to quantify GLY and AMPA in serum showed a strong interference by matrix components, affecting retention time and chromatograms [15,17]. In other studies, although limited to soil samples, fluorenylmethyloxycarbonyl (FMOC) has been used to derivatize GLY, leading to the formation of an adduct absorbing in the UV range [18]. Here, we aimed to develop and validate a simple chromatographic method, using a high-performance liquid chromatography (HPLC) system with a UV detector, for the quantification of GLY and AMPA in different matrices: honey, plasma, water, and flour. The derivatization with 4-Toluenesulfonyl chloride represents a significant advancement in the quantification of GLY and AMPA. While existing methods provide effective solutions, many overlook the importance of simplicity and standardization for routine applications. Our method minimizes the complex steps required by traditional derivatizations, eliminating the need for internal isotopes or advanced instrumentation. This makes it particularly suitable for laboratories operating in resource-limited settings. Furthermore, we demonstrated that our technique is robust and applicable to various matrices, expanding its usability in both academic and industrial contexts.

## 2. Materials and Methods

### 2.1. Chemicals

HPLC-grade acetonitrile (ACN), ethyl-acetate, H_2_SO_4_, and methanol were purchased from VWR Chemicals (Radnor, PA, USA); HPLC-grade water was produced with a Milli-DI system coupled with a Synergy 185 system by Millipore (Milan, Italy). AMPA and GLY were purchased from Sigma–Aldrich Corporation (Milan, Italy). Blank plasma from healthy donors were kindly supplied by the Blood Bank of the San Luigi Gonzaga University Hospital in Orbassano (Turin, Italy). All powders were stored at −20 degrees, in order to prevent any potential degradation.

### 2.2. Stock Solutions, Standards (STDs), and Quality Controls (QCs)

In order to obtain tosylate GLY and AMPA standard solution, the disclosed procedure was carried out:The GLY or AMPA solution was mixed with a tosylchloride solution containing ethyl acetate, obtaining a first aqueous phase and a first organic phase;The obtained aqueous phase was rinsed twice with ethyl acetate and brought to pH 2 by adding 130 µL of 9 M H_2_SO_4_, obtaining a second aqueous phase and a second organic phase;The two organic phases were collected and mixed together, anhydrified, and mixed with ethyl acetate, obtaining a third organic phase;The third organic phase was anhydrified with Mg_2_SO_4_, obtaining a powder;The powder was then suspended in ethyl acetate, obtaining a solution;The solution was dried through nitrogen influx and then transferred for final exsiccation in a vacuum refrigerated concentrator, obtaining a powder;The powder was suspended in 500 µL of methanol and dried again through vacuum refrigerated concentration and subsequently resuspended in ethyl acetate. The standard solutions contained GLY tosylate and AMPA tosylate in a concentration comprised between 0.5 ug/mL and 20.0 ug/mL;Stock solutions were stored at −80 degrees.

The purity of both tosyl-derivatized GLY and AMPA, identified, respectively, as [(4-methylbenzene-1-sulfonyl)(phosphonomethyl)amino]acetic acid and {[(4-methylbenzene-1-sulfonyl)amino]methyl}phosphonic acid, is determined using a GC-MS assay. This involves evaluating the percentage distribution of the peak area corresponding to the analyte in relation to the total peak areas in the chromatogram. The analytical method for performing this test was developed under the conditions described by Kerry-Ann daCosta et al. [19].

Aliquots of the highest STD sample of the calibration curve and QCs were prepared by independently spiking blank plasma with stock solutions, and then stored at −20 °C. The same calibration ranges and QC concentrations were used both for GLY and AMPA: STD 5, 10.0 mg/ L; STD 4, 1.0 mg/L; STD 3, 0.5 mg/L; STD 2, 0.1 mg/L; STD 1, 0.05 mg/L; QC H (high), 5.0 mg/L; QC L (low), 0.5 mg/L.

All procedures (stock solution, STD, and QC preparation and extraction steps) were carried out at room temperature.

### 2.3. Extraction and Quantification in Samples

The following reagents were used for the extraction procedure:Methanol solution 0.005% in trifluoroacetic acid; 2 mL of the obtained solution was cooled in wet ice bath for 20 min;Phosphate buffer 0.4 M (1:1 of Na_2_HPO_4_ 0.4 M and Na_3_PO_4_ 0.4 M, pH 11.00);Derivatizing solution (5 mg of 4-Toluenesulfonyl chloride in 10 mL of ACN).

The extraction procedure was carried out as follows:If the matrix was a cellular or tissue lysate or water, it was deproteinized and delipidized by adding 200 µL of sample and 400 uL of reagent A in a 2 mL tube, and then mixed vigorously for 30 s and centrifuged at 0 °C, 13,000 RPM for 10 min; then, 200 µL of supernatant was transferred to a new 2 mL tube. If the matrix is flour or honey, 100 mg of flour or 100 mg of honey was transferred into a 2 mL tube, 600 µL of reagent A was added, and the procedure was carried out as previously described.In total, 200 µL of sample was transferred to a 2 mL tube, and 200 µL of reagent B and 200 µL of reagent C were added, then vortexed for 10 s.The solution was incubated in a thermostatic bath at 50 °C for 5 min.In total, 500 µL of ethyl acetate was added; then, the obtained solution was mixed vigorously for 5 min and centrifuged at 5 °C, 3500 RPM for 5 min; the supernatant was collected and transferred to a new 2 mL tube. The procedure was repeated with 300 µL of ethyl acetate; the supernatant was washed with 150 µL of toluene in order to eliminate water residues.The supernatant was dried in a vacuum refrigerated concentrator.The dried supernatant was reconstituted in 200 µL of water 0.1%*v*/*v* formic acid and 50 mM ammonium formate; the obtained samples were suitable for HPLC-UV analysis.

### 2.4. UHPLC-UV Analysis

HPLC analysis was performed through the Hitachi Elite LaChrom HPLC System with L-2400 UV (VWR, Radnor, PA, USA) equipped with an autosampler, a spectrophotometer, and a heated column compartment. System management and data acquisition were performed with the EzChrom Elite software (VWR, Radnor, PA, USA). Chromatographic separation was obtained on a Reversed Phase Core Shell (5 µm, 4.6 mm × 100 mm) column (Kinetex, Phenomenex, Torrance, CA, USA).

The mobile phases were the following:

Phase A: Water + 0.1%*v*/*v* formic acid + 50 mM ammonium formate;

Phase B: Acetonitrile–water 9.5:0.5 + 0.1%*v*/*v* formic acid + 50 mM ammonium formate.

The gradient elution was set as follows: at 0.01 min, 90% Phase A and 10% Phase B; at 5 and 7 min, 45% Phase A and 55% Phase B; at 12 and 15 min, 5% Phase A and 95% Phase B; at 16.5 and 18 min, 90% Phase A and 10% Phase B. The analysis was carried out at a constant flow rate of 1 mL/min with an injection volume range of 5.0–50.0 µL. The eluates were monitored at 240 nm. The total runtime was 18 min.

### 2.5. Specificity, Selectivity, Accuracy, Precision, and Limit of Quantification and Detection

The validation of the analytical method to determine AMPA and GLY was implemented in accordance with Commission Decision 2002/657 and Eurachem Guides (The Fitness for Purpose of Analytical Methods: A Laboratory Guide to Method Validation and Related Topics). The upper limit of quantification (ULOQ) corresponded to the highest concentration calibration STD, for both the analytes; the lower limit of quantification (LLOQ) for each analyte was the lowest concentration of analyte in a sample which could be reliably quantified, with a deviation from the nominal concentration, as a measure of accuracy, and relative standard deviation (RSD), as a measure of precision, lower than 20% and with a signal-to-noise ratio higher than 5. The limit of detection (LOD) was considered the lowest dilution of LLOQ, which yielded a signal-to-noise ratio higher than 3. In order to ensure good coverage, the defined calibration range was used to quantify an STD with a higher concentration than the ULOQ, spiked at 10.0 mg/L for both analytes.

### 2.6. Recovery (REC) and Extraction Efficiency (EE)

Percent recovery was obtained evaluating the spike height ratio between the extracted sample and analytes in mobile phase solution at equal concentration (0.05, 0.1, 1.0, 10.0 mg/L for both AMPA and GLY). The final value was obtained as a mean of 10 ratios. The EE was measured by comparing the areas of peaks of pre- and post-spiked samples.

### 2.7. Method Robustness (R) and Matrix Effect (ME)

Method robustness has been assessed by performing pesticide quantification in QCs at three concentration levels (LLOQ, 5.0% ULOQ, 10% ULOQ), in four different matrices: flour, water, human plasma, and honey. Ten replicates of each concentration level have been processed in two independent analytical runs.

The evaluation of interactions between different matrices conditions and the instrumental response was conducted by determining the response factor (RF) calculated as follow:(1)$$RF_{i}=\frac{A_{i}}{C_{i}},$$
where *A_i_* was the single analyte area and *C_i_* was the related concentration. The deviations for QCs in each plasma conditions for each analyte were evaluated as *RF* percent difference (Δ*RF*%) as follows:(2)$$\Delta RF\%=\frac{RF_{i}-\bar{RF}}{\bar{RF}}\times 100$$
where the mean *RF* was calculated by relation(3)$$\bar{RF}=\frac{\sum_{i=1}^{n}RF_{i}}{n}.$$

In the last equation, the *n* = 3 was the number of concentration levels of QCs for each analyte.

To assess statistical deviation between repeatability in normal plasma condition and in haemolytic/lipemic plasma condition, the Fisher test was performed at 97.5 level of confidence. The following equation was applied:(4)$$F=\frac{s_{r1}^{2}}{s_{r2}^{2}}<F_{tab};$$
where $F_{\left(v\;=\;9;\;\alpha \;=\;0.1\right)}=5.39$**,** where *s_r_*_1_ and *s_r_*_2_ represent the standard deviation, respectively, in repeatability normal conditions and in haemolytic/lipemic conditions.

To evaluate precision, the percentage of coefficient variation (CV%) for each analyte in each matrix condition was calculated and the acceptance criterion was ΔCV% ± 20%, as recommended by Eurachem guidelines.

Furthermore, in accordance with Eurachem guidelines, the matrix effect was evaluated by four replicates of QC1, QC2, and QC3. The analytes’ average response was compared to theorical concentrations with a percent deviation in an acceptability range of ±15% for each concentration level above LLOQ and ±20% at LLOQ. The same acceptability values were considered for CV% in precision evaluation of response data in these experimental sessions.

### 2.8. Stability

Stability was assessed by keeping single aliquots of the QCs in the following conditions: 24 h benchtop at room temperature, 24 h at 37 °C, 24 h at 4 °C, 24 h at −20 °C and 1, 2, 4, 5, and 6 months at −80 °C. Three freeze–thaw cycles were monitored. All the tested conditions were compared with freshly extracted QCs. If the measured concentration remained within 15% of the nominal concentration, the analyte was considered stable, in accordance with paragraph 3.2.8. of ICH M10 guidelines [20].

## 3. Results

### 3.1. Specificity and Selectivity

Mean retention times for the considered analytes were 3.16 minutes for AMPA and 5.08 min for GLY (Figure 1). The minor peaks are considered solvent background noise.

### 3.2. Accuracy, Imprecision, ULOQs, LLOQs, and LODs

Accuracy and imprecision values for each analyte at the three QC levels satisfied the Commission Decision 2002/657. For both analytes, the LLOQ/ppm (as parts per million) was 0.05 mg/L, the lower limit of detection (LOD)/ppm value was 0.01 mg/L, and the measure range/ppm was 0.05–10.0 mg/L (Table 1). Calibration curves had a good fit with linear through zero regression models, with a 1/x weighting factor, to ensure high accuracy at low concentrations. Determination coefficients (r2) of calibration curves were all above >0.995. The defined calibration range was 0.05–10.0 µg/mL (see Figure 1).

### 3.3. Recovery (REC) and Extraction Efficiency (EE)

All the parameters satisfied the Commission Decision 2002/657. Mean values were as follows: REC was 80.5% (RSD 5.2%) for AMPA and 85.2% (RSD 5.8%) for GLY; EE was 79.8% (RSD 5.5%) for AMPA and 84.3% (RSD 5.4%) for GLY. The average recovery values refer to the determination performed on three replicates at concentration levels corresponding to QC1, QC2, and QC3, equal to 0.1, 0.5, and 1.0 ppm as theoretical concentrations. The tests were conducted on plasma matrix, water, and cell culture medium, showing a nearly similar trend across all matrices.

### 3.4. Method Robustness (R) and Matrix Effect (ME)

The ΔRF% for all analytes in each plasma condition remained within ±15%; in precision evaluation, the concentration for each analyte was retained within acceptable CV% in each matrix condition; results are reported in Table 2, for GLY and AMPA and referred to a 5.0% ULOQ level. In detail, for AMPA and GLY, matrices showed a greater ΔCV%, in repeatability. Our method was validated according to ICH M10 guidelines, ensuring high standards of robustness and reproducibility. Robustness was tested through the quantification of QCs at three concentration levels across four different matrices (human plasma, water, honey, and flour). Specifically, the method demonstrated a RF% deviation within ±15%, fully meeting acceptability criteria.

No statistical deviation was observed between repeatability in normal plasma condition and in haemolytic/lipemic conditions, with F < 5.39. The CV% for QC2 and QC3 in four different plasma batches was within 20% for all analytes, as requested by guidelines.

The matrix effect data demonstrate that the method effectively mitigates matrix-induced variability. The angular coefficients (m) across the matrices are relatively consistent, with deviations ranging from −8.5% to 4.0%. These results highlight the robustness of the analytical protocol, as the observed deviations remain within acceptable limits, ensuring reliable quantification across diverse matrices such as flour, water, honey, and plasma (see Table 3). In particular, the degradation of GLY to AMPA in the cell culture medium is shown in the chromatogram in Figure 2. The CV% for QC2 and QC3, assessed during the matrix effect precision study, met the established acceptability criteria (see Table 2).

### 3.5. Stability and Incurred Sample Reanalysis (ISR)

No photodegradation was observed for AMPA or GLY. Both tosylated derivatives of GLY and AMPA in methanolic solution at QC concentrations proved to be stable under the tested conditions, with variations in concentration not exceeding 15%. Stability was confirmed for 24 h at 4 °C, −20 °C, and room temperature, as well as for 24 h at 37 °C, and for up to 6 months at −80 °C. As required by the ICH M10 guidelines, samples were re-analyzed to evaluate incurred sample reanalysis (ISR). The results demonstrated acceptable bias, with values of 11.2% for tosylated GLY and 14.2% for tosylated AMPA.

## 4. Discussion

To sum up, the proposed method included the following steps (see scheme reported in Figure 3):Adding to the sample suspected of containing GLY and/or AMPA a tosylchloride solution, therefore obtaining a first solution;Incubating the first solution, thus obtaining GLY-tosylate and AMPA-tosylate;Extracting GLY-tosylate and AMPA-tosylate from the first solution, by mixing the first solution with an organic aprotic solvent having a dielectric constant value lower than 10, centrifuging and collecting a supernatant;Drying the supernatant;Reconstituting the supernatant in ACN, obtaining a reconstituted supernatant;Chromatographically separating GLY-tosylate and AMPA-tosylate from other constituents in the reconstituted supernatant: UV detection of GLY-tosylate and AMPA-tosylate, determining the presence and/or the amount of GLY and AMPA in the sample.

The experiments were carried out in four different matrices: flour, water, honey, and human plasma.

The choice of these matrices was strategic to demonstrate the versatility of the method. The matrices included in the study represent relevant application scenarios: human plasma for toxicological analysis, water for environmental monitoring, honey as a representative of complex foods, and flour for processed food products. This selection covers a range of key applications; however, the method is potentially applicable to many other matrices, such as biological tissues and soils. Further studies could expand the application scope, further strengthening the validity of the method.

Our extraction and quantification procedures are oriented towards UV detection with a widespread application scenario. Several reports refer to the tandem mass technique, which is a second-level analysis (confirmation analysis), while our method can be broadly applied in the quantification of spectrophotometric detectable analytes. This extraction protocol provides for a simpler procedure with an easy determination by UV spectrometry. Not all the analysis laboratories have mass detectors available: the possibility of using simpler and cheaper instrumentation can ensure greater applicability of our developed method. Moreover, our invention clearly overcomes other critical issues often related to GLY and AMPA extraction/quantification in biological matrices, due to the polar nature of the molecule or matrix interference. Several publications available in the literature refer to FMOC derivatization techniques [21,22,23], which is more complex compared to the 4-Toluenesulfonyl chloride derivatization contained in this proposal. Very often, the matrix effect in AMPA detection with tandem mass techniques is mitigated by employed isotope internal standard: this approach is very expensive and does not necessarily lead to reliable readings [4]. Some derivatization protocols based on FMOC lack accuracy when applied to biological matrices, while others are developed for fluorometric detection [23,24,25,26,27].

The proposed GLY and AMPA extraction and quantification are based on a completely novel derivatization procedure and will allow operators to reduce costs and time to perform the assay, improving repeatability and standardization among different labs. The strengths of our method are the high specificity for biological matrices (e.g., toxicological application) and the affordable analytical technique (HPLC-UV) applicable to a vast sample processing capacity.

A higher level of samples and fortified matrix was required for the validation of analytical methods, according to UNI EN ISO/IEC 17025:2018. The performance parameters have been tested with intralaboratory analytical protocols as requested by 7.2.2. of UNI EN ISO IEC 17025:2018.

The evaluated parameters were as follows: Shapiro–Wilk test for range of linearity and normality, lower limit of detection, lower limit of quantification, repeatability, reproducibility and robustness, and the uncertainty of measure evaluation with a metrological approach and the Horwitz heuristic model. We, therefore, obtained a transversal extraction method potentially applicable to all biological samples, with a chemical optimization of the proposed procedure.

Beyond the strictly analytical aspects, which quantitatively highlight the ability of the proposed method to identify and quantify GLY and AMPA with specificity, selectivity, precision, repeatability, reproducibility, robustness, and, where applicable, compensate for the matrix effect, it is essential to consider the instrumental aspects. Often, suggestions, thoughts, and ideas can arise from reading between the lines of history, particularly the history of science, which above all teaches us how to best apply the scientific method: the methodology.

In our case, tosyl derivatives were used for the quantitative analysis of amines and amino acids in the late 1980s, not in liquid chromatography, but rather in gas chromatography coupled with single-quadrupole mass spectrometry (GC-MS) [28]. The goal was not so much to modify the boiling point (which for the tosylated derivative of glyphosate is approximately 260 °C) but to increase its lipophilicity, thereby extending its retention time on traditional polymethylsiloxane capillary columns. This idea, however, suggested to others the possibility of spectroscopically detecting glyphosate and AMPA using UV detection coupled with high-pressure liquid chromatography (HPLC-UV) [14,15]. In this way, hydrophilic, low-molecular-weight compounds, which would otherwise typically require ion chromatography for their determination, could be quantitatively analyzed with spectroscopic detectors.

As mass spectrometry techniques have gradually “gained ground”, enabling coupling with increasingly efficient chromatographic systems (UPLC) [29], today, with a UPLC-MS/MS system, the quantitative determination of glyphosate, AMPA, glufosinate, and many other polar pesticides can be performed on virtually any type of matrix with a chromatographic run of just a few minutes, thanks to new-generation stationary phases such as HILIC or similar [30].

Why then, after almost 40 years, consider a method involving an older system like HPLC-UV? The reasons can be varied. Certainly, the economic sustainability of the instrumentation must be considered first: the cost ratio is approximately 1:10 or even higher when high-end UPLC-MS/MS systems are taken into account. From a more analytical perspective, the motivations have essentially been summarized in the previous arguments.

## 5. Conclusions

GLY-based herbicides have been extensively used worldwide for almost 50 years, and we are facing a continuous exposure to undetermined levels of GLY and AMPA in water, soil, fruits, and food [31]. The effects on human and animal health of GLY and AMPA are still profoundly disputed, and it is beyond the scope of this study [32,33,34,35,36,37,38,39,40]. Nevertheless, the research progress on potential deleterious outcomes needs to be supported by the precise quantification of GLY and AMPA in a wide range of samples, including biological fluids, cellular and tissue specimens, samples obtained from food processing, and others. To date, there is no transversal protocol capable of unifying the extraction and quantification procedures adopted by research and development, quality control, university, and other institutional labs. Our procedure aims to unify protocols to extract and determine GLY and AMPA in different matrices, with UV chromatography. The developed method will represent a practical resource for a wide range of application fields, including experimental, medical, quality control, alimentary, and several others. One of the primary objectives of our study was to ensure that the method could be easily scaled for large-scale applications. The simplicity of the derivatization protocol and the availability of cost-effective reagents make the method ideal for quality and industrial control laboratories. Additionally, the 18 min analysis time per sample is competitive compared to other approaches, allowing for high-throughput sample processing. This aspect is particularly relevant for applications in food safety and environmental monitoring, where operational efficiency is crucial.

## 6. Patents

EP4348269A1, a method to determine glyphosate and aminomethylphosphonic acid in a sample. D.M., S.D.F., E.A., S.A., and F.C. hold the international patent filed on 26 May 2022.

## Figures and Tables

**Figure 1 jox-15-00023-f001:**
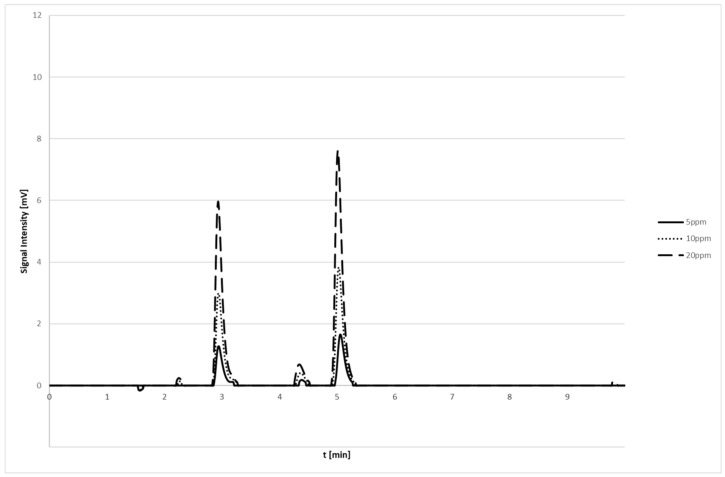
Representative chromatograms of calibration curves: standard solutions with increasing concentrations of glyphosate and AMPA (5, 10, 20 ppm). The first peak refers to AMPA (RT = 3.16 min), while second peak is relative to glyphosate (RT = 5.08 min). The minor peaks are due to solvent background noise.

**Figure 2 jox-15-00023-f002:**
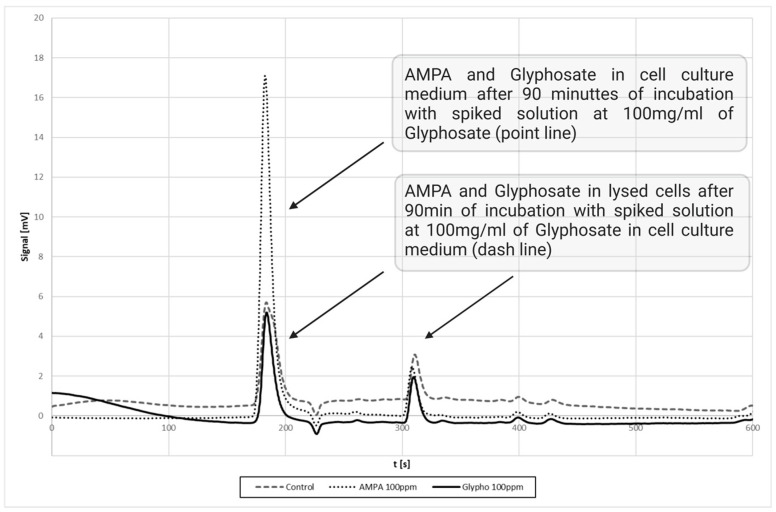
Time course of glyphosate degradation to AMPA. AMPA and glyphosate in cell culture medium after 90 min of incubation with spiked solution at 100 mg/mL of glyphosate (point line); AMPA and glyphosate in lysed cells after 90 min of incubation with spiked solution at 100 mg/mL of glyphosate in cell culture medium (dashed line).

**Figure 3 jox-15-00023-f003:**
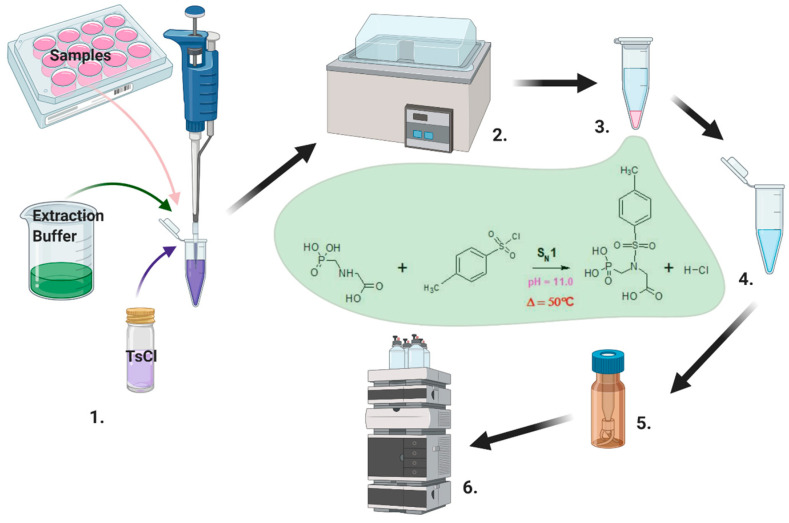
The workflow illustrates the process of sample preparation and analysis. (1) Samples are treated with an extraction buffer and TsCl (p-toluenesulfonyl chloride) to initiate a derivatization reaction, enhancing analyte properties by converting hydroxyl or amino groups into tosyl derivatives. (2) The reaction mixture is incubated at elevated temperatures (e.g., 50 °C) in a controlled environment, followed by (3) centrifugation to separate components. (4) The supernatant is then carefully transferred to a clean tube for further processing. Finally, the processed sample is placed into a vial (5) and analyzed using high-performance liquid chromatography (HPLC) for detailed compound characterization (6).

**Table 1 jox-15-00023-t001:** Retention time of peaks and relative concentration for glyphosate and AMPA.

Analyte	RT/[min]	LOD/ppm	LOQ/ppm	Measure Range/ppm
Glyphosate	5.08	0.01	0.03	0.05–20.0
AMPA	3.16	0.01	0.03	0.05–20.0

**Table 2 jox-15-00023-t002:** Summary of glyphosate quantification results obtained through HPLC-UV analysis after derivatization with tosyl chloride. Data include mean values (xm, mg/kg), residuals (%) relative to nominal concentrations, repeatability standard deviation (sr, mg/kg), reproducibility standard deviation (sR, mg/kg), uncertainty (U, mg/kg) calculated using the Horwitz equation, and coefficient of variation (CV%). Results are presented for four matrices (flour, water, honey, and plasma) at three nominal concentration levels (0.1, 0.5, and 1.0 mg/kg).

Matrix	Residual (%)	Mean Value (xm)	Repeatability Standard Deviation (sr)	Reproducibility Standard Deviation (sR)	Uncertainty (U)	CV%
Flour	10.0	0.11	0.014	0.015	0.3	12.7
	12.0	0.56	0.041	0.058	0.2	7.3
	6.0	1.06	0.103	0.115	0.2	9.7
Water	−10.0	0.09	0.012	0.014	0.3	13.3
	−10.0	0.45	0.039	0.045	0.2	8.7
	−6.0	0.94	0.054	0.067	0.1	5.7
Honey	0.0	0.1	0.015	0.017	0.3	15.0
	0.0	0.5	0.034	0.039	0.2	6.8
	−1.0	0.99	0.064	0.091	0.2	6.5
Plasma	10.0	0.11	0.014	0.014	0.2	12.7
	2.0	0.51	0.041	0.046	0.2	8.0
	3.0	1.03	0.087	0.099	0.2	8.4

**Table 3 jox-15-00023-t003:** Matrix effect data showing the angular coefficient of the matrix calibration curve (m) and the percentage deviation of the matrix calibration slope (Δm%) for each tested matrix.

Matrix	Angular Coefficient of the Matrix Calibration Curve (m)	Matrix Deviation in Percentage (Δm%)
Flour	22,856	−8.5%
Water	25,981	4.0%
Honey	23,640	−5.3%
Plasma	22,931	−8.2%

## Data Availability

No data were used for the research described in the article.

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
