# Peer review of "New Standardized Procedure to Extract Glyphosate and Aminomethylphosphonic Acid from Different Matrices: A Kit for HPLC-UV Detection"

_jox, 2025, doi:10.3390/jox15010023_

Round 1
Reviewer 1 Report
Comments and Suggestions for Authors
The manuscript needs to be implemented in the descriptive part of the results. It is understandable but too brief. The authors have made a brief and little argued description.
Explain well why they prepare the reference standard and at what purity do they obtain it?
The extraction is described for biological samples but they also talk about flour, water and honey. Is the method applied the same? In general, biological samples have different procedures than environmental samples. How were the samples of flour, water and honey extracted?
In the recovery and extraction efficiency paragraph, the recovery values are given but it is not understood to which matrix they refer, how many replicates for each matrix/level combination have been made. Authors should provide such data.
In the paragraph Robustness and matrix effect, no information is given about the matrix effect and matrices specified in the job. Authors should provide such data.
In the stability section, authors should provide information on what the stability study refers to: (1) if to the prepared reference standard, (2) to the compounds analysed in matrices or (3) the AMPA and GLY standards in solution. They should provide experimental data on stability over time.
The discussion paragraph should be argued, it seems a technical report.
Author Response
Dear reviewer,
really thank you for your comments and suggestions.
Following you can find response point by point to second part of your revision:
- Explain well why they prepare the reference standard and at what purity do they obtain it?
The preparation of metrological reference solutions of glyphosate and aminomethylphosphonic acid, which therefore represent secondary analytical standards, is conducted as described step by step in paragraph 2.2, letters a) to h), lines 140 to 157. This type of preparation is necessary because the tosyl derivatives of both glyphosate and aminomethylphosphonic acid are not commercially available. However, we have integrated the paragraph by describing the purity test employed to verify the concentration of the obtained standards for both glyphosate and aminomethylphosphonic acid (see lines 158-164).
- The extraction is described for biological samples but they also talk about flour, water and honey. Is the method applied the same? In general, biological samples have different procedures than environmental samples. How were the samples of flour, water and honey extracted?
This is a good point. In fact, the extraction process in the first part, concerning flour and honey, is slightly different as it accounts for the hygroscopicity and hydration state of the matrix. Therefore, paragraph 2.3, lines 182–184, has been updated accordingly with the following text: If the matrix is flour or honey, 100 mg of flour or 100 mg of honey is transferred into a 2 mL tube, 600 µL of reagent A is added, and the procedure is carried out as previously described.
- In the recovery and extraction efficiency paragraph, the recovery values are given but it is not understood to which matrix they refer, how many replicates for each matrix/level combination have been made. Authors should provide such data.
Thank you for the observation. We have integrated the article (lines 285–289) with the following specifications: The average recovery values refer to the determination performed on three replicates at concentration levels corresponding to QC1, QC2, and QC3, equal to 0.1, 0.5, and 1.0 ppm as theoretical concentrations. The tests were conducted on plasma matrix, water, and cell culture medium, showing a nearly similar trend across all matrices.
- In the paragraph Robustness and matrix effect, no information is given about the matrix effect and matrices specified in the job. Authors should provide such data.
Thank you for pointing this out. We have now included a detailed table (Table 3) reporting the angular coefficients of the matrix calibration curves (m) and the percentage deviations (Δm%) for each tested matrix (flour, water, honey, and plasma). These data demonstrate the method's efficiency in mitigating matrix effects, as the observed deviations range from -8.5% to 4.0%. The analytical protocol has shown to be robust and reliable, ensuring consistent quantification across all tested matrices. The updated information has been added to the manuscript in the Method Robustness (R) and Matrix Effect (ME) paragraph 3.4, lines 303-309.
- In the stability section, authors should provide information on what the stability study refers to: (1) if to the prepared reference standard, (2) to the compounds analysed in matrices or (3) the AMPA and GLY standards in solution. They should provide experimental data on stability over time.
Thank you for your valuable comment. We have clarified the scope of the stability study in the manuscript. Specifically:
- The stability study refers to the prepared reference standards of tosylated GLY and AMPA in methanolic solution at QC concentrations. These standards were tested under various conditions, including 24 hours at 4°C, -20°C, room temperature, and 37°C, as well as up to 6 months at -80°C. The results demonstrated variations within acceptable limits (≤15%) across all tested conditions.
- Additionally, as required by the ICH M10 guidelines, incurred samples were re-analyzed to evaluate incurred sample reanalysis (ISR). The results showed acceptable bias, with values of 11.2% for tosylated GLY and 14.2% for tosylated AMPA.
This additional information has been included in the Stability section of the manuscript (paragraph 2.8 line 269 and paragraph 3.5 lines 311–318).
- The discussion paragraph should be argued, it seems a technical report.
Thank you for your insightful comment. We have revised the discussion section to include a more comprehensive analysis and contextualization of our findings. Specifically, we have expanded the discussion by highlighting both the analytical and instrumental aspects of the proposed method.
In particular, we have elaborated on the historical use of tosyl derivatives for the analysis of GLY and AMPA, providing insights into the evolution of chromatographic and mass spectrometry techniques over the past decades. Additionally, we have discussed the rationale for revisiting an HPLC-UV approach, focusing on economic sustainability, as well as the analytical motivations that make this method a viable alternative to modern UPLC-MS/MS systems.
These additions provide a critical perspective on the relevance and implications of the proposed method, as well as its place within the broader context of analytical chemistry. The updated discussion can be found in lines 402-430 of the manuscript.

Reviewer 2 Report
Comments and Suggestions for Authors
1) Glyphosate is not a new substance, and many studies have been conducted on its detection methods and population risk assessment. The author's statement that there is no toxicological research on it is incorrect;
2) The use of 4-Toluenesulfonyl chloride as a derivatization reagent may be the only innovation in this article, but the background and usage of 4-Toluenesulfonyl chloride are not explained in the introduction;
3) The matrix involved in this article includes honey, plasma, water and flour,Among them, water does not belong to biological matrices;
4) The standard method involves joint validation by multiple laboratories, so please do not mention "standardized";
5) Please provide a large number of references from the past five years;
6) The unit of addition for the recovery rate experiment varies depending on the substrate, please revise it;
7) Please revise the non written writing of PPM;
8) The title and abstract of the paper need to be carefully revised.
Comments on the Quality of English Language1) Glyphosate is not a new substance, and many studies have been conducted on its detection methods and population risk assessment. The author's statement that there is no toxicological research on it is incorrect;
2) The use of 4-Toluenesulfonyl chloride as a derivatization reagent may be the only innovation in this article, but the background and usage of 4-Toluenesulfonyl chloride are not explained in the introduction;
3) The matrix involved in this article includes honey, plasma, water and flour,Among them, water does not belong to biological matrices;
4) The standard method involves joint validation by multiple laboratories, so please do not mention "standardized";
5) Please provide a large number of references from the past five years;
6) The unit of addition for the recovery rate experiment varies depending on the substrate, please revise it;
7) Please revise the non written writing of PPM;
8) The title and abstract of the paper need to be carefully revised.
Author Response
Dear reviewer,
really thank you for your comments and suggestions.
According to your prior part of revision, we can affirm that english has been revised, and that the introduction and conclusion sections have been improved.
Following you can find response point by point to second part of your revision:
1) Glyphosate is not a new substance, and many studies have been conducted on its detection methods and population risk assessment. The author's statement that there is no toxicological research on it is incorrect;
In the introduction section of paper (lines 73-79) toxicological research about glyphosate is indicated. However our aim was to underline that scientific literature about the biological effects of glyphosate on humans is still controversial and still misses consensus.
2) The use of 4-Toluenesulfonyl chloride as a derivatization reagent may be the only innovation in this article, but the background and usage of 4-Toluenesulfonyl chloride are not explained in the introduction;
You are correct, thank you. We added in the References list these papers 10.1016/j.chroma.2005.07.062, 10.1016/0378-4347(91)80130-5, 10.1016/S0021-9673(01)88832-4 and we indicated the discussion of the use of 4-Toluenesulfonyl chloride in the introduction section of paper (lines 95-97 ).
3) The matrix involved in this article includes honey, plasma, water and flour,Among them, water does not belong to biological matrices;
Water for environmental monitoring (as indicated in the text at line 319) should be considered biological matrix.
4) The standard method involves joint validation by multiple laboratories, so please do not mention "standardized";
Standardization of analytical methods, according to UNI EN ISO IEC 17025:2018 does not require interlaboratory validation. However we changed wrong formulated phrase in discussion section (lines 349-352)
5) Please provide a large number of references from the past five years;
Recent scientific literature about glyphosate determination is not abundant. By the way, we added following papers to references list, indicating them also in the text (10.3390/toxics10100552, 10.1016/j.aca.2020.05.058, 10.1016/j.scitotenv.2023.162499)
6) The unit of addition for the recovery rate experiment varies depending on the substrate, please revise it;
Please, could you clarify this comment?
7) Please revise the non written writing of PPM;
At line 266 of the text we indicated for the first time the non written "ppm" as parts per million
8) The title and abstract of the paper need to be carefully revised.
We changed title as following: "New standardized procedure to extract glyphosate and aminomethylphosphonic acid from different matrices: a kit for HPLC-UV detection"
Abstract has been revised carefully.
Abstract
Background
Glyphosate has been extensively used as an herbicide since the early 1970s. The daily exposure limit 16 is set at 0.3 mg/kg bw/d in Europe and 1.75 mg/kg bw/d in the USA. Among its derivatives, aminomethylphosphonic acid is the most stable and abundant. Understanding their biological effects then requires reliable methods for quantification in biological samples.
Methods
We developed and validated a fast, low-cost, and reliable chromatographic method for determining glyphosate and aminomethylphosphonic acid concentrations. The validation included following parameters: specificity, selectivity, matrix effect, accuracy, precision, calibration performance, limit of quantification, recovery, and stability. Sample extraction employed an anion exchange resin with elution using hydrochloric acid 50.0 mmol/L. For HPLC analysis, analytes were derivatized, separated on a C18 column with a mobile phase of phosphate buffer (0.20 mol/L, pH 3.0) and acetonitrile (85:15), and detected at 240 nm.
Results
The method demonstrated high reliability and reproducibility across various matrices. Its performance met all validation criteria, confirming its suitability for quantifying glyphosate and aminomethylphosphonic acid in different biological and experimental setups.
Conclusions
This method can offer a practical resource for applications in experimental research, medical diagnostics, quality control, and food safety.

Round 2
Reviewer 1 Report
Comments and Suggestions for Authors
The paper is updated according to the comments.
Author Response
Thank you for your positive comment to our paper.
Reviewer 2 Report
Comments and Suggestions for Authors
1) Glyphosate is not a new substance, and many studies have been conducted on its detection methods and population risk assessment. The author's statement that there is no toxicological research on it is incorrect; You should change the wording in the article, it's impossible without relevant researchï¼›
3) The matrix involved in this article includes honey, plasma, water and flour,Among them, water does not belong to biological matrices; I disagree with your answer
6) The unit of addition for the recovery rate experiment varies depending on the substrate, please revise it; Solid matrix uses mg/kg, liquid matrix uses mg/L
7) Please revise the non written writing of PPM; Scientific units of measurement should be used,like mg/kg
Author Response
Dear reviewer,
really thank you for your comments and suggestions.
Following you can find response point by point of your revision:
- Glyphosate is not a new substance, and many studies have been conducted on its detection methods and population risk assessment. The author's statement that there is no toxicological research on it is incorrect; You should change the wording in the article, it's impossible without relevant research.
Thank you for your comment. I carefully reviewed the point raised and agree that the original statement might have been inaccurate, given the extensive literature available on glyphosate, both in terms of detection methods and population risk assessment. As a result, I have removed the sentence corresponding to lines 75–76 to align the content with scientific standards and improve the accuracy of the manuscript.
- The matrix involved in this article includes honey, plasma, water and flour,Among them, water does not belong to biological matrices; I disagree with your answer
Thank you for your observation. I have carefully considered your comment and agree with your assessment. As a result, I have removed “biological” term when useful and eliminated this consideration from the manuscript.
6) The unit of addition for the recovery rate experiment varies depending on the substrate, please revise it; Solid matrix uses mg/kg, liquid matrix uses mg/L
Thank you for this point. The relative recovery percentage, by definition, is independent of the unit of measurement used to express the analyte concentration. As outlined in the ICH M10 guidelines, Section 7.3, recovery is defined as the ratio multiplied by 100 of the difference in concentration between the spiked matrix and the blank, relative to the theoretical concentration of the spiked matrix. Being a ratio of concentrations, it is dimensionless by definition and expressed as a percentage.
7) Please revise the non written writing of PPM; Scientific units of measurement should be used,like mg/kg
Thank you for your comment. The use of "ppm" in the manuscript is intentional and aligned with established metrological conventions. Both mg/kg and mg/L are valid units of measurement; however, since the study involves both solid and liquid matrices, we have opted for "ppm," which implicitly represents a mass/volume or mass/mass ratio depending on the context. This approach is consistent with the recommendations outlined in the NIST Special Publication 330 (SP330), which recognizes "ppm" as a dimensionless quantity useful for expressing relative proportions. We believe this choice ensures clarity and consistency across the manuscript.
